# Sentence Representation Method Based on Multi-Layer Semantic Network

**Wenfeng Zheng [1] , Xiangjun Liu [1] and Lirong Yin [2],***

[1] School of Automation, University of Electronic Science and Technology of China, Chengdu 610054, China; winfirms@uestc.edu.cn (W.Z.); ShineDream.Liu@gmail.com (X.L.)

[2] Department of Geography and Anthropology, Louisiana State University, Baton Rouge, LA 70803, USA

* Correspondence: yin.lyra@gmail.com

**Abstract:** With the development of artificial intelligence, more and more people hope that computers can understand human language through natural language technology, learn to think like human beings, and finally replace human beings to complete the highly difficult tasks with cognitive ability. As the key technology of natural language understanding, sentence representation reasoning technology mainly focuses on the sentence representation method and the reasoning model. Although the performance has been improved, there are still some problems such as incomplete sentence semantic expression, lack of depth of reasoning model, and lack of interpretability of the reasoning process. In this paper, a multi-layer semantic representation network is designed for sentence representation. The multi-attention mechanism obtains the semantic information of different levels of a sentence. The word order information of the sentence is also integrated by adding the relative position mask between words to reduce the uncertainty caused by word order. Finally, the method is verified on the task of text implication recognition and emotion classification. The experimental results show that the multi-layer semantic representation network can promote sentence representation's accuracy and comprehensiveness.

**Keywords:** semantics; sentence representation; natural language reasoning; multi-layer network; multi-attention mechanism

## 1. Introduction

Natural language inference (NLI) has become one of the most important benchmark tasks in the field of natural language understanding because of its complex language understanding and in-depth information involved in reasoning [1]. Natural language reasoning technology is widely used in automatic reasoning [2], machine translation [3], question answering systems [4], and large-scale content analysis. Compared with computer vision and speech recognition technology, natural language reasoning has not reached a high level because of its technical difficulties and complex application scenarios [5,6]. Once the natural language reasoning technology makes a breakthrough and realizes the real barrier-free communication between humans and machines, human life quality will be greatly improved.

The sentence representation method's research focuses on improving the sentence representation module's performance to obtain complete and accurate semantic information coding to improve the method's performance. The sentence representation module aims to map natural language sentences into a dense vector space under the premise of keeping the sentence expression semantics unchanged. It transforms the complex logical reasoning process into the solution process of similarity between sentences and solving the relationship between sentences by computer [7–9].

Before the emergence of sentence-level representation technology, sentence semantic representation used Continuous Bag of Words (CBOW) embedded distributed representation technology based on word coding to represent the text as a fixed-length sentence vector.

Although this method can calculate sentence representation succinctly and efficiently, it loses the sequence information between words. Different sentences will get the same representation regardless of the word order if they use the same word. Even if we consider the word order in a short time (the predicted context range is greater than 1), sentence representation is also affected by data sparsity and high dimensions. For example, words such as "Paris", "France", and "capital" are equidistant in embedded representation, but "Paris" is closer to "Capital" in semantics.

Faruqui [10] proposed an unsupervised method for characterizing the sentence paragraph vector learning continuous distributed variable text vector representation. By joining together the sentence embedded vector and other embedded vector of each word to predict the probable next word, the variable text can be phrases, sentences, or document. Kiros [11] extracts from specific tasks and expands the distributed semantic assumption of words in the skip-gram model [10,12]. It is assumed that sentences with similar context information usually have the same or similar semantic information. Based on this assumption, a general sentence representation model, skip-thought, for learning high-quality sentence vectors is proposed. Unlike paragraph vector, the skip-thought model uses words to predict their context. It encodes a complete sentence to predict the sentences around it, thus obtaining sentence representation. The skip-thought model requires that the training set's text be sequential, meaning that the three sentences input must be sequential. It presents a problem for migration to social media or artificial languages generated by symbolic knowledge. To solve this problem, Hill [13] proposed Sequential. Denoising Auto-Encoder (SDAE). SDAE uses the deep network model built layer by layer with an automatic noise reduction encoder to automatically learn useful semantic features from many unsupervised data. It then uses the extracted semantic features to predict source sentences. The biggest difference from skip-thought is that SDAE can train sentence sets in any order. Based on the distribution hypothesis at the sentence level [14], Hill [13] proposed another sentence representation method FastSent. It uses cosine distance to directly query the representation space, simply adds the same signals, and realizes the improvement of convergence speed by simplifying the skip-thought model.

With neural networks and deep learning development, sentence representation technology has gradually developed from combining the simple word embedding model and more complex neural network architecture. For example, convolution networks [15,16], cyclic neural networks [17], and their variants [18,19] have been applied to improve the performance of sentence representation.

Zhao [20] proposed an adjacent convolutional reasoning model by taking advantage of the convolutional neural network's advantages in feature extraction. Zhao [20] spliced all the extracted semantic features to form the input text's embedded semantic representation by extracting semantic features of different abstract levels of text. Although the neural network is good at capturing the training model's deviation, it is easy to forget the input statement's complete information. It is too focused on reducing the deviation, which leads to the poor effect of the final model on the test set. Liu [21] combined the monolayer self-attention mechanism based on graves. Chen [22] combined the depth threshold attention mechanism to improve characterization performance. The existing attention mechanism only focuses on a certain level of sentence information. It cannot fully express the meaning of the sentence. Therefore, Shen [23] proposed a multi-head attention mechanism that uses eight layers of 600-dimensional attentions. Each layer has 75 hidden neurons, which are used to encode sentence semantics. Based on these studies, Shen [24] designed a bidirectional self-attention network Disan. It uses a self-attention mechanism to encode sentence semantic scalar and uses multi-layer attention to expand the obtained semantic scalar to vector expression. This method's advantage is that it does not need to use a neural network to code sentence semantics to avoid falling into the local optimum. The disadvantage is that it pays too much attention to semantic information. Too little attention is paid to the sentence's word order information, which easily leads to inaccurate sentence semantic expression.

This paper designed a sentence representation method based on multi-layer semantics, analyzed the importance of sentence semantics for sentence representation, and learned the key technologies of sentence representation. The existing sentence representation methods cannot correctly and comprehensively express the meaning of a sentence. This paper uses the combination of bidirectional long-term and short-term memory networks and a multi-attention mechanism to obtain the semantic information of different levels of the sentence. The relative position information between words is added into the sentence word order information. Finally, all the information is fused to form a complete sentence embedded representation.

## 2. Dataset

### 2.1. SNLI Dataset

The SNLI data set is a text implication recognition data set published by Stanford University. SNLI is manually annotated and contains 570 k text pairs, used as testing and training sets for NLI systems [25–28]. There are three kinds of marks: implication, contradiction, and neutral. In this paper, all data are divided into a training set (549,367 samples), a verification set (9842 samples), and a test set (9824 samples), according to Zhu's [29] data partition rules. Example SNLI data forms are shown in Table 1.

**Table 1.** Sample data of SNLI dataset.

| Premise Sentence | The Label | Hypothetical Sentence |
| --- | --- | --- |
| Two women are embracing while holding to go packages. | Entailment E E E E E | Two woman are holding packages. |
| A man selling donuts to a customer during a world exhibition event held in the city of Angeles. | Contradiction C C C C C | A woman drinks her coffee in a small café. |
| A man in a blue shirt standing in front of a garage-like structure painted with geometric designs. | Neutral N E N N N | A man is repainting a garage. |

### 2.2. Multi-NLI Dataset

The Multi-NLI dataset [30] contains 433 k text pairs, which is different from the SNLI dataset [31]. It covers more data close to real life, such as novels and telephone voice. The sample data is shown in Table 2. The data set contains 10 categories of data. According to whether the same category appears in the training set and test set simultaneously, it is divided into matched and mismatched set.

**Table 2.** Sample data of Multi-NLI dataset.

| Category | Premise Sentence | The Label | Hypothetical Sentence |
| --- | --- | --- | --- |
| Novel | The Old One always comforted Ca'daan, except today. | neutral | Ca'daan knew the Old One very well. |
| Message | Your gift is appreciated by each and every student who will benefit from your generosity. | neutral | Hundreds of students will benefit from your generosity. |
| Cell | yes now you know if everybody like in August when everybody's on vacation or something we can dress a little more casual or | contradiction | August is a black out month for vacations in the company. |

The text implication task is carried out on the matching set and the unmatched set. The data are divided into a training set (392,702 samples) and a matching/mismatching verification set (9815/9832 samples). Since the test set data cannot be obtained, this paper uses the verification set instead of a test set.

### 2.3. Yelp Dataset

Yelp data set is the public data provided by Yelp, the largest review website in the United States. It is mainly used for recommendation systems and sentiment analysis tasks. It provides the forms of JSON and SQL, which can be directly used in any application. This paper randomly selects 500,000 samples from the Yelp dataset as the model's training set, 2000 samples as the verification set, and 2000 samples as the test set.

## 3. Methods

Natural language semantics refers to the probability and meaning contained in natural language sentences. Semantics express the meaning of words and covers various logical relationships such as causality and transition between things. In short, semantics is the description and logical representation of things [32,33].

### 3.1. Semantic Extraction Based on Bidirectional Long-Short Memory Network

The core of the sentence representation method is the design of the representation network. Sentence representation based on the neural network has gradually become the main method in sentence representation. Although the recurrent neural network can retain the information between adjacent words, it cannot learn the information between the words far away in the sentence. The emergence of long–short term memory (LSTM) solves the problem that the recurrent neural networks cannot learn long-term information. The LSTM and RNN both have the chain shape of repetitive neural network modules. The difference is that the unit modules are different. As shown in Figure 1, three gates are added to the LSTM unit model: input gate, forgetting gate, and output gate.

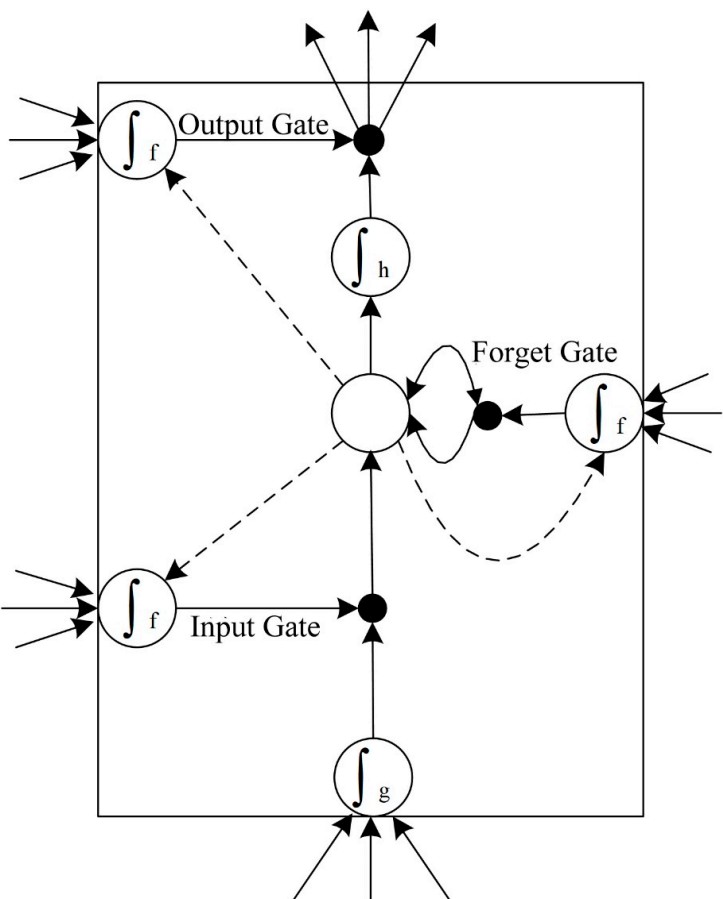

**Figure 1.** Schematic diagram of long short-term memory network unit module.

LSTM first determines what information should be discarded at time *t* through the forgetting gate, then determines what information should be saved at time *t* through the input layer, then updates the temporary information according to the information of input gate and forgetting gate, and finally outputs the result of time t and the hidden layer state $h_t$. The hidden layer state $h_t$ will serve as input to the next neuron. Suppose that the input at time *t* is $x_t$. The temporary information is $C_t$. The output is $o_t$. The calculation formula of each door is shown in Formulas (1)–(6).

(1)　Forget gate

$$f_t = \sigma\left(W_f[h_{t-1}, x_t] + b_f\right) \tag{1}$$

(2)　Input gate

$$i_t = \sigma(W_i[h_{t-1}, x_t] + b_i) \tag{2}$$

$$\widetilde{C}_t = tanh(W_C[h_{t-1}, x_t] + b_C) \tag{3}$$

(3)　Output gate

$$C_t = f_t * C_{t-1} + i_t * \widetilde{C}_t \tag{4}$$

$$o_t = \sigma(W_o[h_{t-1}, x_t] + b_o) \tag{5}$$

$$h_t = o_t * tanh(C_t) \tag{6}$$

Among them, $W_f, W_i, W_o,$ and $W_C$ are the weight parameter of each gate in the unit module; B is the weight parameter of each gate in the unit module; $b_f, b_i, b_o,$ and $b_C$ are the corresponding bias variables; $\sigma$ and *tanh* are the activation functions of the network; $h_{t-1}$ and $C_{t-1}$ were the state of hidden layer and cell at the last moment; *and* $\widetilde{C_t}$ represents the information that should be updated in the cell at time *t*.

The disadvantage of LSTM in sentence representation modeling is that it only considers single direction word order when more fine-grained classification is needed.

Bi-directional long-short memory network (BiLSTM) considers the hidden layer state $h_t = [h_{Lt}, h_{Rt}]$ of two directions of LSTM at the same time and solves the problem of bidirectional semantic dependency. The structure of BiLSTM is shown in Figure 2. The forward and backward network structures can be consistent or different from each other.

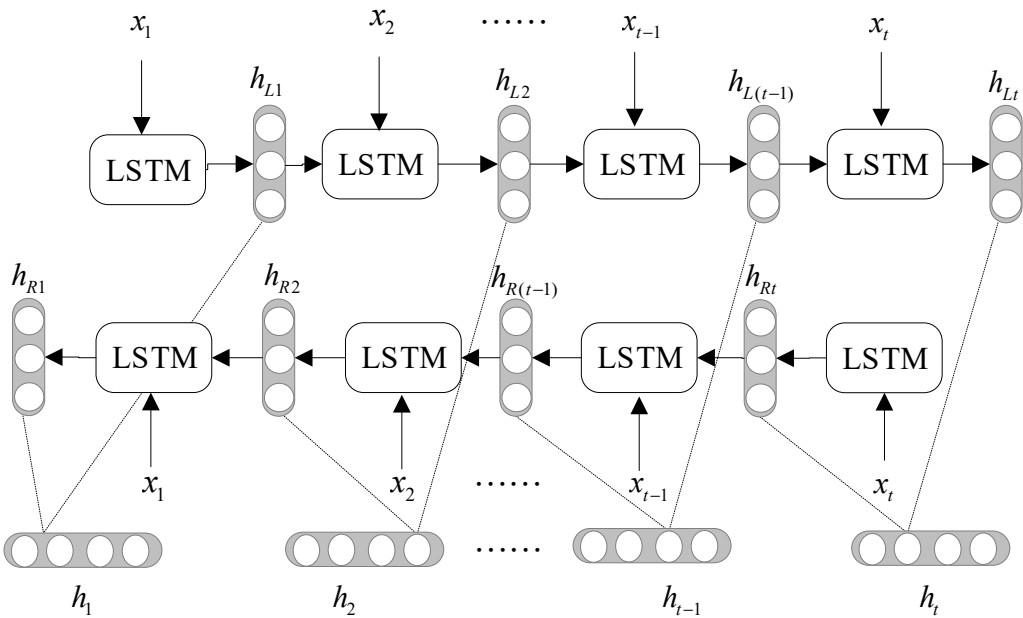

**Figure 2.** Operation diagram of the bidirectional long-short memory network.

### 3.2. Design of Semantic Representation Network Based on Multi Attention

The multi-layer semantic representation network includes three steps: sentence vectorization, semantic information extraction, and reinforcement, and sentence embedding representation. Specifically, the two-way long-term and short-term memory networks are used as the basic framework of the network. The multi-attention mechanism is used to construct a multi-layer semantic representation network to obtain the semantic information of different levels of sentences. The relative position information is added to the network to strengthen the core semantic relationship in the sentence and reduce the interference of redundant information.

#### 3.2.1. Natural Language Sentence Vectorization

The function of sentence vectorization is to obtain the initial embedded representation of the sentence and the semantic representation network's embedding layer. This part is composed of the Seq2Word module, Word2Char module, and embedded layer. The network structure is shown in Figure 3.

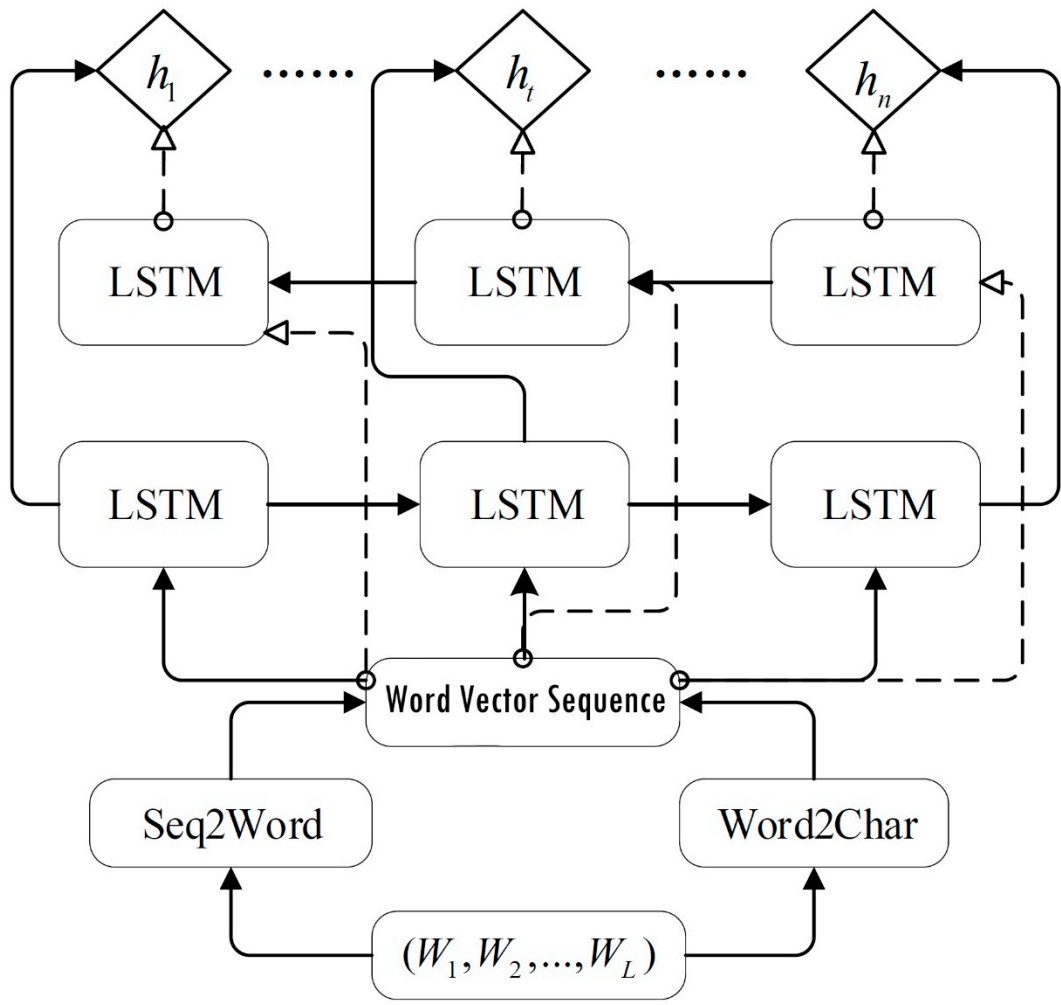

**Figure 3.** Network structure diagram of the embedded layer.

Suppose input $S = (W_1, W_i, \ldots, W_L)$, depending on the word $W_i$ in sentence $S$ input into the embedding layer of multi-layer semantic representation network, and calculate the initial embedding representation $H$ of sentence $S$, where $W_i$ ($i = 1, 2, \ldots, L$) is the $i$th word of sentence $S$, and $L$ is the length of sentence. The calculation process of each module is as follows:

1. Seq2Word module

The module is mainly used to obtain the embedded representation of the word vector corresponding to each word in the sentence $S$, the word-level word vectorization processing. The sentence is mapped from the natural language space to the vector space by splicing all the word vector representations of the sentence. The specific steps are as follows:

(1) Word2vec technology [10] combined with GloVe-840B-300D [34] was used to train the word vector embedding representation of the dictionary database;

(2) According to the word's position in the dictionary, the embedding vector w corresponding to each word in the sentence is obtained $w_i \in R^{d_w \times 1}$.

2. Word2Char module

The module uses convolution networks of different sizes to extract character features in words [35] and obtain word embedding representation at the character level. The specific steps are as follows:

(1) According to the given character dictionary (including 24 letters and common punctuation and special characters), the word $W_i$ is split into a list of characters;

(2) After a single-layer convolution network and pooling layer, the character vector $c_i \in R^{m \times d_c}$ of the word $W_i$ is obtained, where $d_c$ is the dimension of each letter mapping, and $m$ is the number of pooling layers. The convolution layer uses $n * h$ convolution kernel ($n$ is the dimension of character embedding vector, $h$ is the size of convolution kernel window).

3. Embedded layer

Firstly, the word vector $w_i$ obtained from the Seq2Word module and the character vector $c_i$ obtained from the Word2Char module are combined to form the semantic vector $e_i \in R^{d_e}$ of the word $w_i$. Finally, the word vector sequence $x = \{e_1, \cdots, e_t, \cdots, e_L\}$ of sentence $S$, where $x \in R^{L \times d_e}$, $d_e$ is the dimension of the semantic vector $e_i$, and $d_e$ is the sum of $d_w$ and $d_c$.

The $N$-layer BiLSTM network is used to obtain further information of the word vector sequence $x$ of the sentence to obtain the context information in the sentence. For each layer of the network, the hidden layer state of the forward network and the backward network $\overrightarrow{h}_t$ and $\overleftarrow{h}_t$, are as follows:

$$\overrightarrow{h}_t = LSTM(e_t, \overrightarrow{h}_{t-1}), \ \overleftarrow{h}_t = LSTM(e_t, \overleftarrow{h}_{t-1}) \tag{7}$$

splicing $\overrightarrow{h}_t$ and $\overleftarrow{h}_t$ to obtain the hidden layer state $h_t = \left[\overrightarrow{h}_t, \overleftarrow{h}_t\right] \in R^{1 \times 2u}$, where $u$ is the number of neurons in the forward (backward) network of each layer. After $N$-layer cycle, the hidden layer state of the top-level network is output as the embedded representation $H \in R^{N \times 2u}$ of sentence $S$.

### 3.2.2. Multi-Layer Semantic Information Extraction and Enhancement

This part is the core of the multi-layer semantic representation network. Its purpose is to obtain as many layers of sentence semantic information as possible. As shown in Figure 4, this paper constructs a multi-layer attention network using the multi-head attention mechanism. It emphasizes the sentence word order by adding a location mask. Finally, the fused multi-layer semantic vector is used as the input of the representation layer.

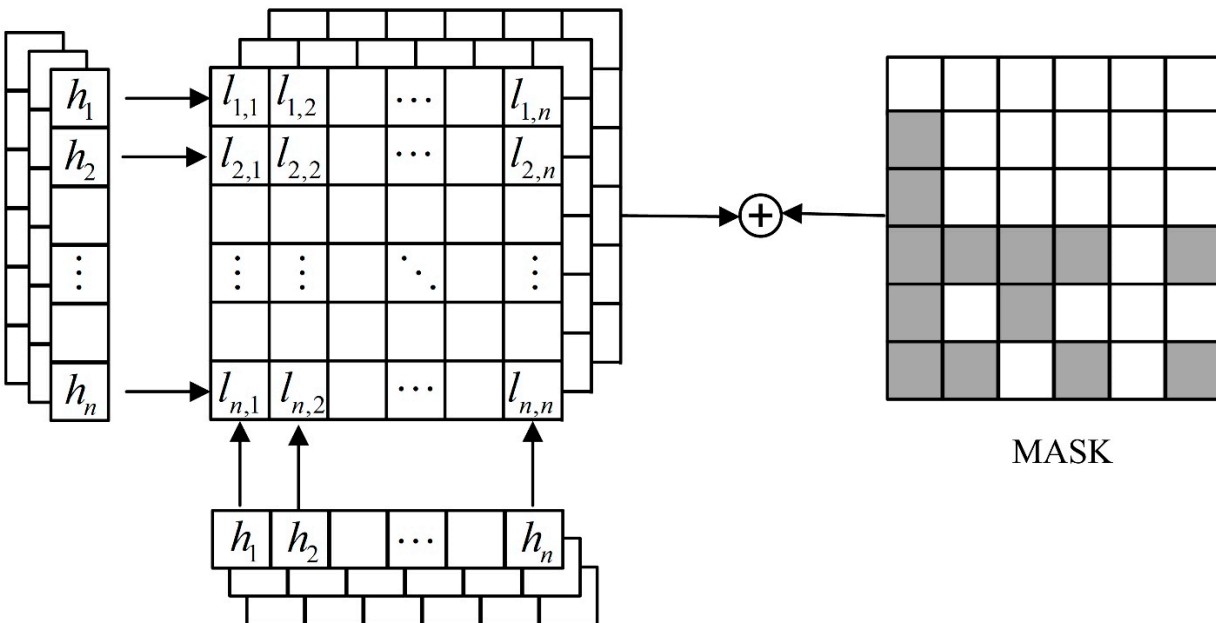

**Figure 4.** Multi-layer semantically encoded network structure diagram.

1. Multi-layer semantic information extraction

The single-layer attention network can only capture the semantic information of a single layer. Influenced by Ling [36], we build a semantic extraction network based on the multi-attention mechanism to capture more semantic information. For the layer $\tau$ in the multilayer attention network, the attention weight vector $a_\tau \in R^{N \times 2u}$ of the layer is calculated first; the calculation method of attention weight vector $a_\tau$ is shown in Formula (8).

$$a_\tau = softmax(\alpha_\tau \sigma(\beta_\tau H + \overline{\alpha}_\tau) + \overline{\beta}_\tau)^T \tag{8}$$

where $\tau = 1, 2, \cdots, \lambda$, $\lambda$ denotes the number of layers of multilayer attention network, $\alpha_\tau \in R^{d_a \times 2u}$, $\beta_\tau \in R^{2u \times d_a}$, $\overline{\alpha}_\tau \in R^{d_a \times 1}$, $\overline{\beta}_\tau \in R^{2u \times 1}$ is the parameter of layer $\tau$, $T$ is a transposition, $\sigma$ is activation function, and $H \in R^{N \times 2u}$ is the embedded representation of sentence $S$.

Although the attention mechanism can capture important semantic information in a sentence, the attention mechanism's calculation method to obtain semantic information is based on the word bag model. It does not pay attention to each word's order. Suppose the self-attention layer mainly captures the phrase components in the sentence. As long as the sentence phrases still have strong internal relations, the sentence is disordered; the results obtained after ordering are consistent with those obtained before the disorder. However, based on the attention weight vector, there are great differences in the understanding of sentences. Therefore, word order $a_\tau$, information is added to strengthen the core semantic relationship of sentences.

2. Word order information enhancement

Word order information can be a deterministic function of position [37] or obtained by network representation learning. This paper obtains word order information by calculating the relative distance between words considering the network structure to not increase the network complexity. Firstly, the absolute distance $D_{ij} = |i - j|$ of any two words in the word vector sequence $x$ corresponding to sentence $S$ is calculated, where $i, j = 1, 2, \ldots, L$, $L$ are the number of participles contained in the sentence.

Then, for the distance, $D_{ij}$ carries on the deviation standardization processing, obtains the word order information $P_{ij}$, ensuring that even in complex sentences (containing more

words) $D_{ij}$ will not calculate the model too large and cause the calculation to be too slow. The calculation formula of $P_{ij}$ is as follows:

$$P_{ij} = \frac{D_{ij} - D_{min}}{Dmin_{max}} \tag{9}$$

where $D_{max}$ is the maximum absolute distance between any two words, and $D_{min}$ is the minimum absolute distance between any two words.

Then, the word order information is used to weight the attention weight vector $a_\tau$, that is, Formula (9) is replaced into Formula (8) to calculate the weighted semantic weight $a_\tau'$, The calculation formula of $a_\tau'$ is shown in (10).

$$a_\tau' = softmax((\alpha_\tau \sigma(\beta_\tau H + \overline{\alpha}_\tau) + \overline{\beta}\tau) \cdot alpha \cdot P_{ij})^T \tag{10}$$

where $\tau = 1, 2, \cdots, \lambda$, $\lambda$ denotes the number of layers of multilayer attention network, and $\alpha_\tau, \beta_\tau, \overline{\alpha}_\tau, \overline{\beta}_\tau$ is the parameter of the $\tau$ layer in the multi-layer semantic network. The parameter $alpha$ regulates the influence of word order information on hierarchical semantic information, adjusted according to specific tasks.

Finally, the multi-layer semantic weight vector $A' \in R^{\lambda \times N \times 2u}$ is formed by splicing the semantic information extracted from the multi-layer attention network

$$A' = concat(a_1', a_2', \ldots, a_\lambda') \tag{11}$$

where $concat(\cdot)$ is the splicing operation.

### 3.2.3. Sentence Embedding Representation Generation

Using the multi-layer semantic weight $A'$, obtained from the multi-layer attention network, the sentence embedding representation $H$ obtained by the embedding layer is weighted to obtain the multi-layer semantic information $M'$, as shown in Formula (12).

$$M' = A' \odot H \tag{12}$$

where $\odot$ denotes point multiplication.

Due to the difference in sentence length, the length of multi-layer semantic information obtained is also different. For sentence representation, the final representation result should be transferable. The size of the sentence embedded representation needs to be unified before the representation result is output. In this paper, through the maximum pooling operation, the sentence embedding representation $V$ of the representation layer is output, as shown below.

$$V = Maxpooling(M') \tag{13}$$

where $Maxpooling(\cdot)$ is the pooling operation.

## 4. Experiment

Experiments are carried out on the text implication task and emotion classification task, compared with the existing sentence representation methods to verify the proposed representation method's performance based on the multi-layer semantic network.

### 4.1. Experimental Steps

For the task of text implication recognition, to avoid the interference caused by the reasoning process, to judge the performance of a multi-layer semantic network, a complete sentence representation reasoning model is formed by combining the general reasoning model [38] and multi-layer semantic representation network. The inference module uses a simple heuristic matching method. The inference information includes the embedded representation of the premise sentence and the hypothetical sentence, the difference between the premise sentence and the hypothetical sentence, and the product of the two.

The prediction module comprises a fully connected neural network and a classifier used to predict and classify the implication relationship. The specific experimental steps are as follows:

(1) Firstly, the multi-layer semantic representation network is used to obtain the sentence representation of the premise statement and the hypothetical sentence as $u$ and $v$;
(2) Then, the inference information $|u - v|$ and $u * v$ between sentences are calculated by the inference module;
(3) Then, the inference information $|u - v|$, $u * v$, premise sentence representation $u$, and hypothetical sentence representation $v$ are spliced and input into the fully connected network [26] to predict the classification implication relationship.

This paper uses Yelp data to do an emotion classification task [29]. According to their comments, this paper judges users' star ratings. It divides them into one star to five stars, from negative evaluation to positive evaluation. The basic experimental steps are the same as above. The only difference is that the final three classifications are replaced by the five.

*4.2. Evaluation Index and Parameter Setting*

4.2.1. Evaluation Index

The evaluation of sentence representation in this paper is mainly divided into the objective evaluation and subjective evaluation. The higher the correlation, the closer the representation content is to the original meaning of the sentence.

The objective evaluation uses the accuracy index to evaluate the performance of the model. The formula of accuracy is as follows:

$$Accuracy = \frac{N_t}{N_{all}} \tag{14}$$

where $N_t$ is the number of samples with a correct prediction, $N_p$ is the number of samples consistent with the target labels in the prediction results, and the total number of samples is $N_{all}$.

4.2.2. Parameter Setting

This paper's experiments are based on the Theano1.0 deep learning platform, using NVIDIA GeForce GTX 1070 hardware acceleration. The general model parameter settings refer to the work of Zhu [29], and the specific parameters are as follows:

(1) Word vector pre-training is initialized by GloVe-840B-300D [34]. For words not in this table, Gaussian distribution is used for random initial vectorization. The final word vector dimension is 300D.
(2) The dimension of the character vector in the Word2Vec model is set to 15. CNN network adopts a single-layer network with 100 channels, and the length of each layer of the network is 1/3/5.
(3) To prevent overfitting, we set Dropout [39] to select from (0.5, 0.65, 0.7, 0.75, 0.8) according to the task to achieve the best effect, and early stop strategy is adopted.
(4) The BiLSTM network dimension of the model on the SNLI dataset is 600. The number of batch learning is set to 32. The initial learning rate is 0.0004; the BiLSTM network dimension of Multi-NLI and Yelp data sets is 300, the number of batch learning is set to 8. The initial learning rate is 0.0002 and 0.0001, respectively.
(5) The model training adopts Adam [40] optimization algorithm.

## 5. Results and Discussion

*5.1. Results*

5.1.1. Experimental Result on SNLI Dataset

Table 3 shows different SNLI data models' performance, including the neural network-based model, attention-mechanism-based model, and sentence representation method proposed in this paper.

**Table 3.** Accuracy of various models in SNLI dataset.

| Model | The Training Set (%) | The Test Set (%) |
|---|---|---|
| 300D LSTM (Bowman et al., 2016) | 83.9 | 80.6 |
| 600D BiLSTM Intra-attention (Liu et al., 2016) | 84.5 | 84.2 |
| 600D BiLSTM Deep-Gated-attention (Chen et al., 2017) | 90.5 | 85.5 |
| BiLSTM (Graves et al., 2013) | 90.4 | 85.0 |
| Multi-head (Vaswani et al., 2017) | 89.6 | 84.2 |
| 300D DiSAN (Shen et.al., 2017) | 91.1 | 85.6 |
| Model in this paper | 91.7 | 86.1 |

The models based on attention mechanism include (1) Multi-head [23]: using only an eight-layer multi-head attention mechanism; (2) DiSAN [24]: 300-dimensional bidirectional self-attention network, including forward and backward self-attention network. The models based on the neural network include (1) 300D LSTM [41]: composed of a 300-dimensional unidirectional LSTM network; (2) BiLSTM [42]: composed of a 600-dimensional BiLSTM, including 300-dimensional forward LSTM and 300-dimensional backward LSTM; (3) BiLSTM Intra-attention [41]: combining a single layer self-attention mechanism based on BiLSTM network; and (4) BiLSTM Deep-Gated attention [22]: Based on the BiLSTM network, it combines the depth threshold attention mechanism.

The experimental results show that the multi-layer semantic representation network's accuracy rate in this paper reaches 91.7% on the SNLI training set and 86.1% on the test set. The highest accuracy rate of the other two models is only 85.6%. The model's performance in this chapter is better than that of the other two types of models. It shows that the accuracy of sentence representation can be effectively improved; by adding multi-layer semantics in the representation, the reasoning result is improved.

### 5.1.2. Experimental Results on Multi-NLI Dataset

Table 4 shows the performance of each model on the Multi-NLI dataset. The CBOW model and BiLSTM model [30] use simple single-layer semantics to generate sentence embedding representation. The accuracy rates of the CBOW model and BiLSTM model [30] are 64.8% and 66.9% on the matching test set, and 64.5% and 66.9% on the unmatched test set, respectively.

**Table 4.** Accuracy of each model on Multi-NLI dataset.

| Model | Matched Set (%) | Unmatched Set (%) |
|---|---|---|
| CBOW (Williams et al., 2017) | 64.8 | 64.5 |
| BiLSTM (Williams et al., 2017) | 66.9 | 66.9 |
| Shorted stacked BiLSTM (Nie et al., 2017) | 74.6 | 73.6 |
| Model in this paper | 73.6 | 73.8 |

Compared with the other three models, the proposed multi-layer semantic representation network's accuracy is 73.6% on the Multi-NLI matching set and 73.8% on the non-matching set. The proposed method's accuracy is 0.2% higher than the best one at present. This result indicates that this paper's sentence representation method is more conducive to capturing complex semantic relations.

### 5.1.3. Experimental Results on Yelp Dataset

The performance of each model on the Yelp dataset is shown in Table 5. The baseline model adopts the neural network structure (BiLSTM/CNN) and pooling layer (maximum pooling/self-attention pooling). Among them, the model of BiLSTM combined with self-attention pooling proposed by Zhu [29] performs best, and the accuracy rate on the Yelp test set reaches 64.2%. In contrast, the accuracy rate of the multi-layer semantic representation

network proposed in this paper is 63.8% on the Yelp test set. However, it does not exceed Lin's model, which also has relatively high accuracy.

**Table 5.** The accuracy of each model in emotion classification task on Yelp dataset.

| Model | The Test Set (%) |
| --- | --- |
| BiLSTM + Max Pooling (Lin et al., 2017) | 61.9 |
| CNN + Max Pooling (Lin et al., 2017) | 62.0 |
| BiLSTM + Self-attention Pooling (Lin et al., 2017) | 64.2 |
| Model in this paper | 63.8 |

The reason for this situation is that, compared with the general classification problem, emotional analysis needs to be divided into several types according to the author's attitude. The various types cannot be completely distinguished. For example, "very good" and "good" have an inclusive relationship and cannot be completely separated. Therefore, this requires more in-depth mining of potential emotions in sentence representation sensory level information. Although the model proposed in this paper can obtain more information and make up for the lack of single-layer attention, the model's depth is not enough to capture deeper emotional analysis tasks. The accuracy rate on emotional tasks has not been improved.

5.1.4. Semantic Relevance Analysis

In addition to quantitative analysis of the model, to test whether the model can capture multi-layer semantic information, qualitative analysis, namely semantic correlation analysis, is carried out. Figures 5 and 6 show the results of visualizing the semantic relevance of the sentence "a person is training his horse for a competition" by using the BiLSTM model [42] and our sentence representation method based on the multi-layer semantic network.

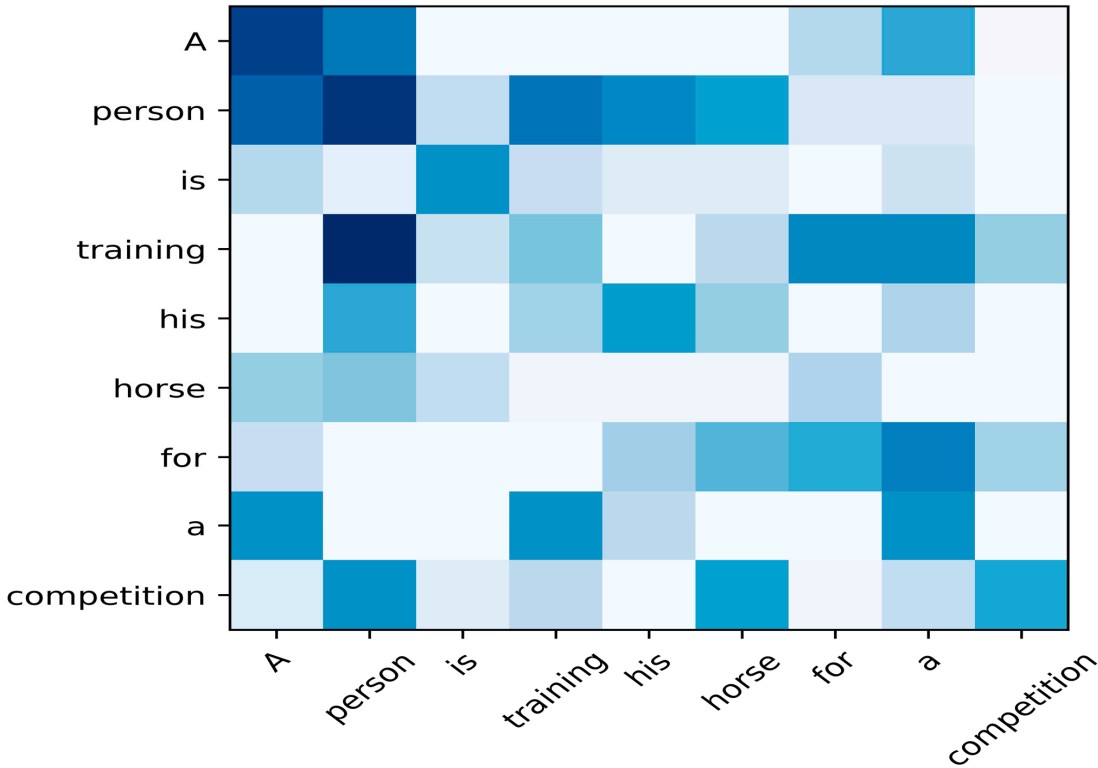

**Figure 5.** Semantic visualization results of the BiLSTM model.

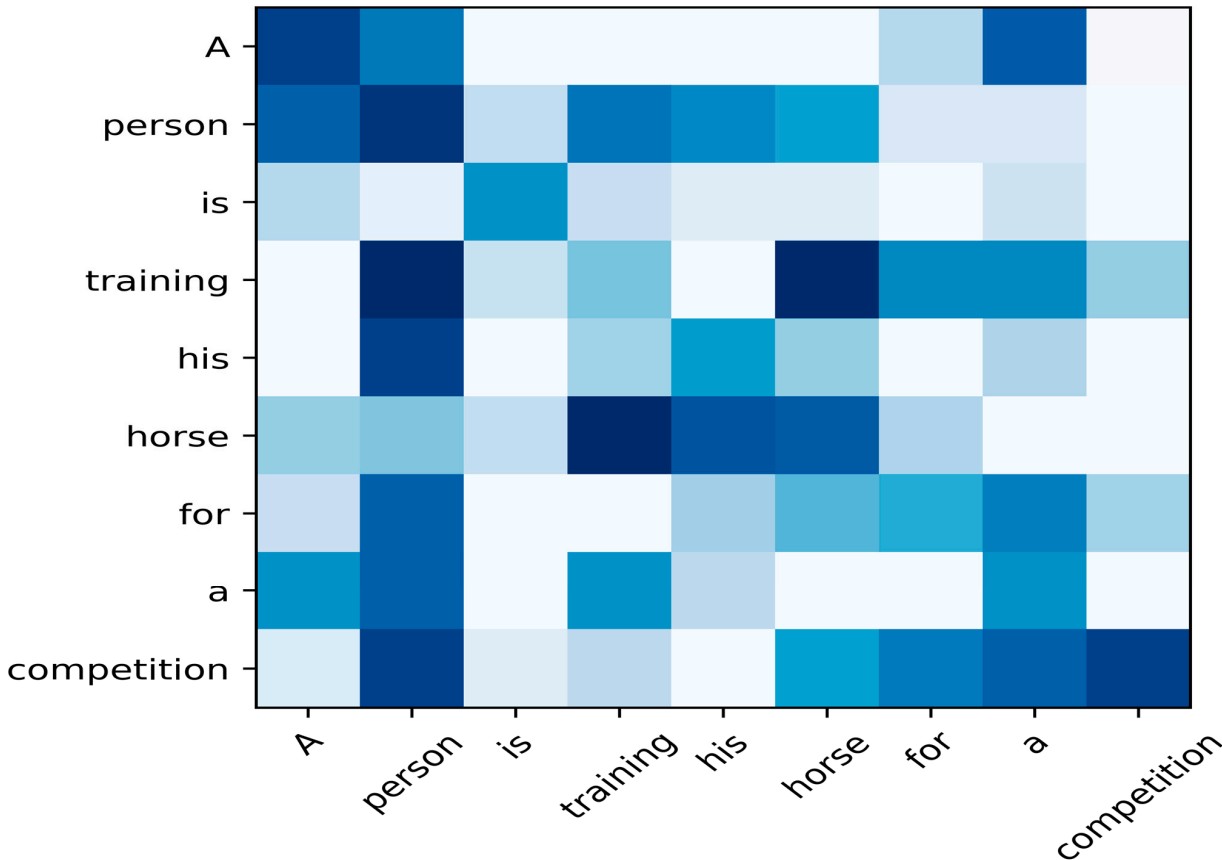

**Figure 6.** Visualization results of multi-layer semantic representation network.

The darker color indicates that the correlation between the two words is more important to the representation of the sentence. Compared with Figures 5 and 6, the multi-layer semantic representation network not only focuses on the relationship between "a person", but also pays attention to other levels of information, such as "trialing his horse" and "for a competition" due to the existence of the multi-attention mechanism. This shows that the sentence representation method based on the multi-layer semantic network can express more comprehensive and accurate sentence information.

### 5.2. Ablation Analysis

Two ablation experiments were designed to explore semantic level and word order information on the representation effect to explore each module's influence in the semantic representation network on the sentence representation method's effect.

### 5.2.1. The Influence of Semantic Levels

In this chapter, semantic layer ablation experiments are carried out on the SNLI dataset for different layers of multi-layer semantic networks. The experimental results are shown in Figure 7.

Figure 7 shows the change of the representation network's accuracy rate with different semantic layers on the SNLI test set. The alignment of different examples in the figure corresponds to a different number of semantic layers (vectors). When the number of semantic layers is less than five, the SNLI data set's accuracy increases with the number of layers. When the number of semantic layers exceeds five, with the SNLI data set as shown in the curve with a vector equal to 7 or 9 in the figure, the accuracy decreases with the increase of the number of layers.

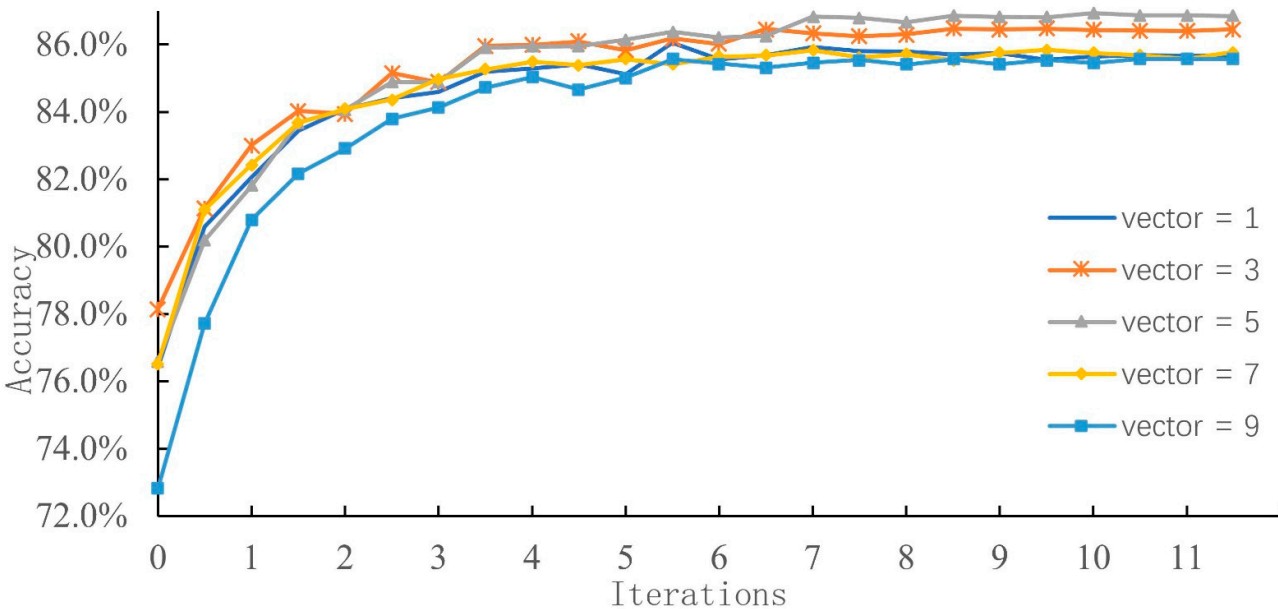

**Figure 7.** Accuracy of different semantic levels on SNLI dataset.

To sum up, too many or too few layers of the multi-layer semantic network may result in the model's poor effect. They cannot improve the sentence representation ability. This situation is that different semantic levels represent the specific semantic information (word meaning, word order, phrase) of a sentence. Too few semantic levels will lead to the loss of important semantics in obtaining sentence semantics. On the contrary, too many semantic levels may lead to redundancy of semantic information of sentences, both of which will have a negative impact on sentence representation.

### 5.2.2. The Influence of Word Order Information

The parameter alpha is the proportion of word order information $P_{ij}$ in the multi-layer semantic network. Figure 8 shows the learning curve of the sentence representation method in this chapter with alpha change on the SNLI dataset. When the number of iterations is equal to 1, the smaller the alpha value, the higher the corresponding SNLI test data set's accuracy. With the increase of the number of iterations, the model's overall learning rate curve first increases and decreases with the increase of alpha. When the learning curve tends to be stable, and the alpha value is equal to 1.5, the model's performance on the SNLI test set is the best.

From the above analysis, we can see that alpha has a great influence on sentence representation. When alpha is equal to 2, the accuracy rate is 85.4%, which is 0.7% lower than the best performance of 86.1%. It also shows that the word order information is of great significance to the semantic representation.

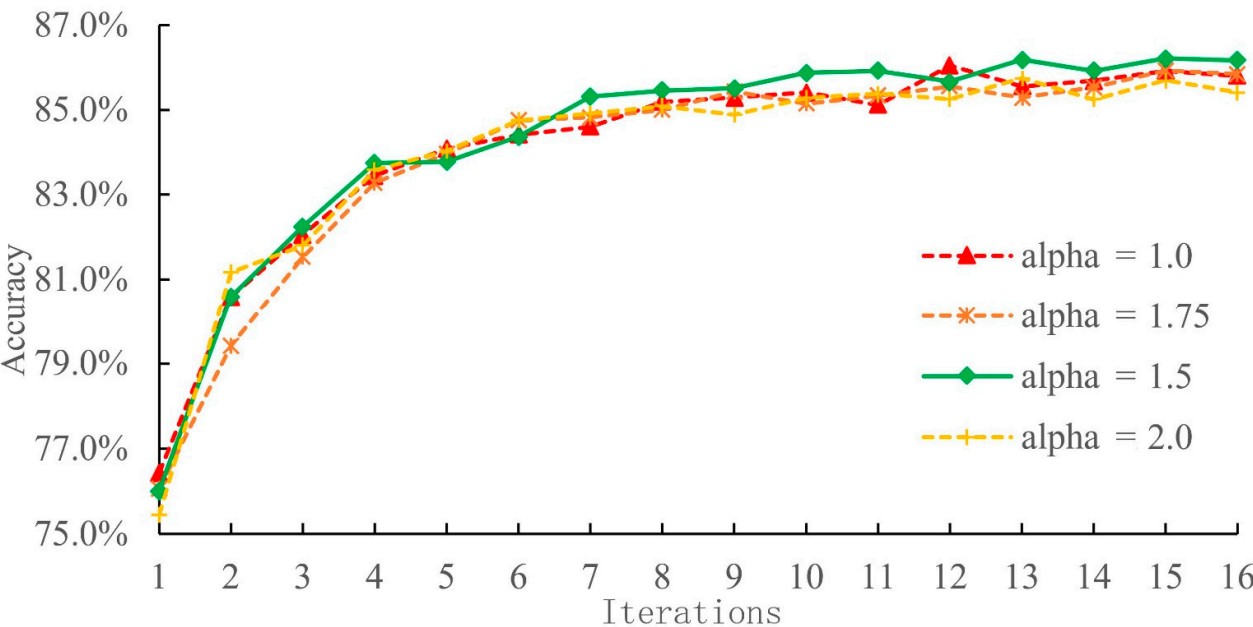

**Figure 8.** The learning curve of different alpha.

## 6. Conclusions

This paper analyzes the characteristics of sentence semantics firstly and then lists the key technologies of existing sentence representation methods. A semantic representation network based on a multi-attention mechanism is designed for sentence representation by summarizing the advantages and disadvantages of existing technologies. This method obtains the semantic information of the same sentence at different levels through a multiple attention mechanism, increases the semantic representation of the sentence, and at the same time, by adding the relative position mask between words, integrates the word order information of the sentence to reduce the uncertainty brought by the word order. Finally, quantitative and qualitative experiments are carried out on the text implication recognition task and emotion classification task. The experimental results show that compared with some traditional networks, the multi-layer semantic representation network designed in this paper significantly improved accuracy on both the SNLI data set and the multi-NLI data set. The model can promote the accuracy and comprehensiveness of sentence meaning representation. However, the Yelp data set's accuracy did not improve significantly, possibly because the model was not deep enough to capture deeper information for emotion analysis tasks.

Although this paper explores the method of sentence representation and reasoning from sentence representation, which improves the reasoning accuracy to some extent, it is far from achieving the best effect. Because of this, future research work can be further studied from the following contents:

In this paper, sentence representation uses multiple levels of semantic and word order information to represent the meaning of sentences. In future research work, the redundant relationship between multiple semantic information levels can be considered to deal. This paper, to extract the semantic information of multiple levels by joining together to form sentences embedded representation directly, does not consider semantic information redundancy between the relationship. Suppose the multilayer extraction between semantic meanings is similar, not only conducive to enhancing the meaning of the sentence comprehensiveness, accumulated in redundant information. In that case, this will also reduce the proportion of the core semantic information, resulting in decreased sentence accuracy.

**Author Contributions:** Conceptualization, W.Z. and L.Y.; formal analysis, X.L.; investigation, X.L.; writing—original draft preparation, L.Y.; writing—review and editing, L.Y.; funding acquisition, W.Z. All authors have read and agreed to the published version of the manuscript.

**Funding:** This work was jointly supported by the Sichuan Science and Technology Program (2021YFQ0003, 2019YJ0189) and the Fundamental Research Funds for the Central Universities (ZYGX2019J059).

**Institutional Review Board Statement:** Not applicable.

**Informed Consent Statement:** Not applicable.

**Data Availability Statement:** Not applicable.

**Conflicts of Interest:** The authors declare no conflict of interest.

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
