# Peer review of "Sentence Representation Method Based on Multi-Layer Semantic Network"

_applsci, doi:10.3390/app11031316_

Round 1

Reviewer 1 Report

This paper is poorly written. There are some English grammar and spelling that needs to be corrected in the introduction ( Example, see lines 60, 61, 62, 73, 74, 448, etc). 

This paper was very difficult to follow. For example, 

Continuous Bag of Words, CBOW is not defined in the paper.

 Stacked denoising Auto-Encoder, SDAE is not defined in the paper.

More structure is required to enhance the flow and understanding of this paper. Perhaps use some flow charts in the introduction to make it easier for the reader to understand the methodology/framework for the research undertaken.

Author Response

  1. The grammar and spelling are refined.
  2. The abbreviations and their explanations are double-checked. All the abbreviations are explained when they first appear in the paper.
  3. The structure of the paper is refined. However, due to the requirement of the journal, the order and number of sections are restricted.
  4. The workflow of this paper is presented in the method parts as flowcharts.

Reviewer 2 Report

Dear authors,

Thank you for your paper. Please find below some comments for the manuscript:

-Abbreviations should be explicitly presented the first time they appear in the text, so that the reader knows what you refer to (e.g. CBOW).

-It is necessary to check the reference style of the journal, as it does not follow it completely.

-The quality of some figures (e.g. Fig. 1, Fig. 4) should be better.

-In section 4.2.2, the title should be in capital letters.

-Kindly check the captions of figures.

-Section 5 is called "Results and discussion", but there is no discussion section.

-References should be revised.

Author Response

  1. The abbreviations and their explanations are double-checked. All the abbreviations are explained when they first appear in the paper.
  2. The reference style is double-checked, and it follows the MDPI style as the template.
  3. The figure quality is refined.
  4. The titles and the captions are rechecked.
  5. The discussion of the results is directly after the presenting of the result. The pros and cons and the possible reasons behind them are discussed in this section.
  6. The reference is refined and revised.

Reviewer 3 Report

This article is showing a Sentence representation method based on the multi-layer semantic network.

Paper is not blinded.

Although the abstract is good, it fails in presenting tangible results.

Section 1 must be divided in two. The introduction should briefly introduce the paper and the respective topics and presenting the work (tasks and goals) and paper structure. New section 2 should explain all the article topics and similar works.

Authors need to improve it.

Section 2

Before presenting the dataset, the authors need to explain their workflow and the methodologies and tools used. Which research method have you used? How was it applied in your work? This section is then complemented by current section 3.

Regarding the dataset, it has not a statistical analysis. Authors need to present a depth analysis of the dataset containing relative and absolute values.

Section 3

The practical methods used are well detailed, although some of the content mentioned before is still missing.

Section 4 and section 5

The practical phase needs to have more details. A set of required details is missing. Authors should take a look at crisp-dm to understand what they should put here.

The ETL process is not well explained. There is not a time frame of the dataset.

This section should present the algorithms used, experiments performed, scenarios created, and models induced. Dataset division is not transparent. Have you used 10folds CV with stratification? How have you ensured a correct division of the dataset?

Why have you presented the results of training and test phases? It is not common to show both.

The measures chosen are also not clear. How was the accuracy achieved? Are you using Confusion matrix? Why have you chosen accuracy and not a set of criteria? Have you defined a threshold to select the best models?

Section 6

This section should have a final critical analysis of the work.

Values are not sustaining the assumptions.

There is no future work or explanation of how this work can be applied in the real world.

The impact of the work on the community or scientific contribution is also not detailed.

Global

The work is interesting; however, the level of details provided are reduced.

A lot of critical questions still not answered.

I cannot assess the quality of the work without having a global overview of it.

Authors need to significantly improve the article and turn the presentation of the work more transparent.

I feel some miss of some scientific rigour.

Authors should avoid writing in the first person.

In my opinion, the paper is incomplete and cannot be accepted in the present form. I advise the authors to improve the article and then submit it to another round of reviews.

Author Response

  1. Due to the requirement of the journal, the order and number of sections are restricted. The introduction cannot be divided into two sections.
  2. Due to the requirement of the journal, the order and number of sections are restricted.
  3. The workflow is explained and presented in the method subsection.
  4. The datasets used in this paper are already well-organized and arranged datasets for the purpose of training and evaluating the new sentence representation method.
  5. Because of the well-organized nature of the dataset, the division is just random selection.
  6. The nature of this paper is presenting a new sentence representation method rather than active data mining. Therefore, the CRISP-DM may not be proper for this paper.
  7. The new method is evaluated and compared with other methods using the same criteria which is the accuracy rate.
  8. Since this paper is focused on presenting a new mixing method instead of a real-world application, the future work is addressed as limitations of this current method and the possible improvement could be further experimented with.

Round 2

Reviewer 3 Report

Authors addressed some of my concerns; however, some minor improvements are still missing.
The idea is not to have a division in the introduction, but having a background section to present the topics and related works.
My advice was to add more details about the work by seeing what CRISP-DM indicates and not converting it in a Data Mining Work.
Authors must prove it "The datasets used in this paper are already well-organized and arranged datasets for training and evaluating the new sentence representation method." and add a statistic analysis of the dataset.
Tangible results are not used in abstract and conclusion.

Author Response

  1. The background and related works are demonstrated in the introduction part because they are supporting the research goal and also because the requirement of formate has restricted names and amount of sections in the paper.
  2. The indicators of the datasets and their quality are not presented in this paper but it is shown in the reference paper of the selected datasets. The quality of this paper also is demonstrated by the past works and related studies that are performed on these datasets.
  3. The detailed statistics and parameters of the datasets could be found in "A large annotated corpus for learning natural language inference" and "The SNLI Corpus".